# Cortical Layer Markers Expression and Increased Synaptic Density in Interstitial Neurons of the White Matter from Drug-Resistant Epilepsy Patients

**DOI:** 10.3390/brainsci13040626

**Published:** 2023-04-06

**Authors:** Jiachao Yang, Mi Wang, Yang Lv, Jiadong Chen

**Affiliations:** 1Department of Neurobiology and Department of Neurology of the Second Affiliated Hospital, Zhejiang University School of Medicine, Hangzhou 310058, China; 18835126261@163.com (J.Y.); 21818574@zju.edu.cn (M.W.); iammeow@163.com (Y.L.); 2NHC and CAMS Key Laboratory of Medical Neurobiology, MOE Frontier Science Center for Brain Research and Brain-Machine Integration, School of Brain Science and Brian Medicine, Zhejiang University, Hangzhou 310058, China

**Keywords:** interstitial neuron, drug-resistant epilepsy, cortical neuron marker, synaptic puncta, white matter

## Abstract

The interstitial neurons in the white matter of the human and non-human primate cortex share a similar developmental origin with subplate neurons and deep-layer cortical neurons. A subset of interstitial neurons expresses the molecular markers of subplate neurons, but whether interstitial neurons express cortical layer markers in the adult human brain remains unexplored. Here we report the expression of cortical layer markers in interstitial neurons in the white matter of the adult human brain, supporting the hypothesis that interstitial neurons could be derived from cortical progenitor cells. Furthermore, we found increased non-phosphorylated neurofilament protein (NPNFP) expression in interstitial neurons in the white matter of drug-resistant epilepsy patients. We also identified the expression of glutamatergic and g-aminobutyric acid (GABAergic) synaptic puncta that were distributed in the perikarya and dendrites of interstitial neurons. The density of glutamatergic and GABAergic synaptic puncta was increased in interstitial neurons in the white matter of drug-resistant epilepsy patients compared with control brain tissues with no history of epilepsy. Together, our results provide important insights of the molecular identity of interstitial neurons in the adult human white matter. Increased synaptic density of interstitial neurons could result in an imbalanced synaptic network in the white matter and participate as part of the epileptic network in drug-resistant epilepsy.

## 1. Introduction

Interstitial neurons, originally described by Ramon y Cajal as neurons situated in the cerebellar white matter, are widely distributed in the cortical white matter of the adult human and non-human primate brain [1,2,3]. Interstitial neurons were considered a small and functionally insignificant population of neurons [4]. The highest density of interstitial neurons is in the white matter immediately adjacent to the gray matter, and this density decreases as the distance from the gray matter increases [5,6,7]. The morphology of interstitial neurons exhibits various features including bipolar or triangular pyramidal-like neurons and multipolar subplate neurons [5,8]. Embryonic 3H-thymidine birth-dating analysis in the monkey or carnivore brain demonstrates that interstitial neurons are generated at the same developmental stage with the genesis of deep-layer cortical neurons or subplate neurons [2,9]. The subplate region is a transient structure during embryonic development, and subplate neurons that survive in the adult brain become superficial (cerebral gyrus) interstitial neurons in the white matter or in layer VIb [2,9,10,11]. The interstitial neurons express various molecular markers, including γ-aminobutyric acid (GABA), microtubule associated protein 2 (MAP2), calbindin, somatostatin (SST), neuropeptide Y (NPY), nitric oxide synthase (NOS), or acetylcholine esterase (AChE), that are also expressed in subplate neurons [4,7,8,12]. However, although a significant portion of interstitial neurons are glutamatergic pyramidal neurons, it remains unexplored whether interstitial neurons could express cortical layer markers [7,13].

Morphological analysis by Golgi staining revealed the presence of spines in the dendritic trees of interstitial neurons that could extend to the deep layer of the cortex [2]. Both symmetric and asymmetric synaptic structures are identified in the perikarya and dendrites of interstitial neurons by electron microscopy or immunohistochemistry, suggesting the presence of both glutamatergic and GABAergic synapses in interstitial neurons in the white matter [2,7,14]. Malformations of cortical development are common causes of drug-resistant epilepsy that are usually accompanied by disordered cortical stratification and indistinct gray and white matter boundaries [15]. In the present study, we analyzed the molecular markers and synaptic density of interstitial neurons in the white matter of drug-resistant epilepsy patients as well as control brain samples with no history of epilepsy. Our results provide important insights into the molecular identity of interstitial neurons and their potential contribution to the epileptic network in drug-resistant epilepsy.

## 2. Materials and Methods

### 2.1. Human Sample Collection

We obtained brain tissues from surgical resections of the epileptogenic focus of drug-resistant epilepsy patients at the Second Affiliated Hospital of Zhejiang University. The study was conducted in accordance with the principles of the Declaration of Helsinki, and the protocol for the handling and use of human tissue was approved by the Medical Ethical Committee of the Zhejiang University School of Medicine (project identification code: Yan2021-0476). The region for resection was defined according to integrated information, including anatomical structures revealed by multimodal preoperative evaluations such as video electroencephalography, magnetic resonance imaging (MRI), or ^18^F-fluoro-deoxyglucose positron emission tomography (PET), and intraoperative electrocorticography. Dysplastic tissue was identified by cortical malformation in acute vibratome slice sections, and dysmorphic neurons were identified by increased soma size and malorientation compared with surrounding cells. Brain tissue were collected from drug-resistant epilepsy patients diagnosed with focal cortical dysplasia (FCD), temporal lobe epilepsy (TLE), and control neocortical brain samples with no history of epilepsy from the Brain Bank at the Zhejiang University School of Medicine, China (National Health and Disease Human Brain Tissue Resource Center). The pathology of FCD type IIB is characterized by the distribution of dysmorphic neurons in the gray matter and balloon cells in the white matter of the epileptic focus. The brain sample information is listed in detail in Table 1.

### 2.2. Immunohistochemical Staining

The human brain tissue was fixed in 4% paraformaldehyde and dehydrated in 30% sucrose at 4 °C for 24–48 h, then cut into thin brain tissue sections (10–20 μm thickness) on a cryomicrotome (Thermo Scientific, NX50, Waltham, MA, USA) and preserved at −80 °C. The immunostaining protocol included the following steps. First, the sections were dried at 60 °C for 10 min, rehydrated in 0.01 M phosphate-buffered saline (PBS), and immersed in sodium citrate buffer for 25 min for antigen retrieval. Second, the sections were air-cooled to room temperature and rinsed three times with 0.01 M PBS for 10 min each. Third, the sections were incubated with 5% donkey serum for 1 h at room temperature in order to block non-specific antibody binding and were subsequently incubated overnight at 4 °C with the following primary antibodies: NeuN (chicken, 1:500, ABN91, Millipore, Darmstadt, HE, German), SMI311 (mouse, 1:500, 837801, BioLegend, San Diego, CA, USA), VGAT (rabbit, 1:500, 131003, Synaptic Systems, Goettingen, SE, German), VGLUT1 (guinea pig, 1:500, AB5905, Millipore, Darmstadt, HE, German), Satb2 (rabbit, 1:500, ab34735, Abcam, Shanghai, China), Ctip2 (rat, 1:500, ab18465, Abcam, Shanghai, China), and Tle4 (mouse, 1:500, sc-365406, Santa Cruz, Dallas, TX, USA). Forth, after primary antibody incubation, the slices were rinsed 4 times in 0.1 mol/L PBS-T (0.1% Triton X-100 in PBS) for 30 min at each time (Triton X-100, Sigma-Aldrich, St. Louis, MO, USA), incubated with fluorescein-conjugated secondary antibodies for 1 h at room temperature, and subsequently washed 4 times in 0.1 mol/L PBS-T for 30 min at each time. The secondary antibodies were as follows: donkey anti-mouse Alexa Fluor 647 (1:1000, A31571, Invitrogen, Shanghai, China), donkey anti-chicken Alexa Fluor 488 (1:1000, 703–545-155, Jackson, West Grove, PA, USA), donkey anti-rabbit Alexa Fluor 555 (1:1000, A31572, ThermoFisher, Waltham, MA, USA), Cy™3 AffiniPure donkey anti-guinea pig (1:1000, 706-165-148, Jackson, West Grove, PA, USA). Fifth, the sections were washed in situ with freshly prepared 70% alcohol for 5 min and incubated with 0.3% Sudan black (0.3 g in 100 mL 70% alcohol) for 15 min in order to reduce the lipofuscin autofluorescence. Finally, the sections were washed 3 times in 0.1 mol/L PBS-T for 10 min each and mounted on glass coverslips using FluorSaveTM reagent (345789, Millipore, Darmstadt, HE, German) and 4’, 6-diamidino-2-phenylindole (DAPI).

### 2.3. Image Acquisition and Data Analysis

Fluorescent images of brain tissue slices were acquired by a confocal laser scanning microscope (FLUOVIEW IX83-FV3000, Olympus, Tokyo, Japan). A whole-mount image of NeuN staining was obtained using a 1.25× dry immersion objective (0.04 NA). Tile scan images were acquired with 20 objective dry immersion objective (0.75 NA, pixel size: 0.621 μm/pixel) or 40 objective dry immersion objective (1.30 NA, pixel size: 0.311 μm/pixel). Confocal images of synaptic puncta were acquired with 60 oil immersion objective (1.42 NA, pixel size: 1× zoom image 0.207 μm/pixel; 3× zoom image 0.069 μm/pixel). Frame size: 1024 × 1024 pixel, dwell time: 4.0 μs/pixel. Multi-track acquisition was images were acquired with excitation lines at 405 nm for DAPI, 488 nm for Alexa 488, 561 nm for Alexa 555, and 640 nm for Alexa 647. Respective emission filters were band-pass 430–470 nm, 500–540 nm, 570–620 nm, and 650–750 nm. For markers’ co-localization analyses, single-plane confocal images were acquired. The immunostaining protocol and image acquisition parameters of the same antibody were kept consistent when imaging across different brain samples.

Tile scan images were analyzed in ImageJ. The numbers of interstitial neurons expressing different cortical layer markers and NPNFP-positive cells were counted from tile scan images (20×) in a region of interest (ROI, 636.4 × 1000 μm) using ImageJ software (1.54d) (National Institutes of Health) and Imaris 9.7.1 software (Oxford Instruments Group, London, England). The border between gray matter and white matter was determined by the distribution of NeuN-positive neurons and DAPI counterstain. The numbers of VGAT- and VGLUT1-positive synaptic puncta were quantified from an ROI (100 × 100 μm^2^) image of each brain sample in the superficial (200–500 μm from the border between gray and white matter) or deep white matter (500–1000 μm from the border between gray and white matter), respectively. The neuron surface area was measured by a cell outline sketch of the soma region labeled by NeuN immunostaining and area measurement in ImageJ software. The number of synaptic puncta in the peri-soma region of interstitial neurons was quantified by averaging the number of synaptic puncta divided by the neuron surface area (per 100 μm^2^) from at least 6 interstitial neurons from each brain sample. Statistical analysis was performed using an unpaired *t*-test, one-way ANOVA, Pearson’s correlation coefficients analysis, and multiple group comparisons using Tukey’s multiple comparisons test on GraphPad Prism 7 (GraphPad Software, San Diego, CA, USA). * *p* < 0.05; ** *p* < 0.01; *** *p* < 0.001; **** *p* < 0.0001. Data are presented as the mean ± standard error of the mean (SEM). 

## 3. Results

### 3.1. Cortical Layer Marker Expression in Interstitial Neurons of the White Matter from the Adult Human Brain

In order to examine the distribution of interstitial neurons in the white matter of adult human brain tissues, we performed immunohistochemistry staining of canonical cortical upper-layer marker special AT-rich sequence-binding protein 2 (Satb2) and deep-layer markers COUP-TF-interacting protein 2 (Ctip2) and transducin-like enhancer protein 4 (Tle4) that reliably labeled cortical neurons in brain tissue (Figure 1A–D and Appendix A) [16,17,18,19,20]. It is important to note the expression of cortical layer markers in interstitial neurons in the white matter of both drug-resistant epilepsy patients and control brain samples with no history of epilepsy. However, we found no differences in the number of neuronal nuclei antigen (NeuN)-expressing interstitial neurons between drug-resistant epilepsy patients with focal cortical dysplasia (FCD) or temporal lobe epilepsy (TLE) and control brain sample, quantified by dividing the white matter every 200 μm bin from the border between the white matter and grey matter to the deep white matter (Figure 1B–E). In addition, the percentage of cortical layer marker Satb2-expressing interstitial neurons (Control: 34.23 ± 11.79%, *n* = 6; FCD: 28.86 ± 3.17%, *p* = 0.9036, *n* = 3; TLE: 21.46 ± 9.66%, *p* = 0.6991, *n* = 3; one-way ANOVA with Tukey’s multiple comparisons test) and deep-layer marker Ctip2 (Control: 67.73 ± 11.48%, *n* = 6; FCD: 62.01 ± 4.75%, *p* = 0.9771, *n* = 3; TLE: 39.84 ± 7.02%, *p* = 0.3522, *n* = 3; one-way ANOVA with Tukey’s multiple comparisons test) or Tle4-expressing interstitial neurons (Control: 12.50 ± 12.50%, *n* = 3; FCD: 40.23 ± 5.31%, *p* = 0.1014, *n* = 3; TLE: 44.72 ± 7.14%, *p* = 0.0622, *n* = 3; one-way ANOVA with Tukey’s multiple comparisons test) in the white matter showed no differences between brain tissues from the drug-resistant epilepsy patients and control brain samples with no history of epilepsy (Figure 1F). These results revealed the presence of interstitial neurons expressing cortical neuronal markers in the white matter of the adult human brain, suggesting a potential origin of interstitial neurons from pallial cortical progenitor cells in the human brain.

### 3.2. Increased Non-Phosphorylated Neurofilament Protein (NPNFP) Expression in Interstitial Neurons in the White Matter of Drug-Resistant Epilepsy Patients

Non-phosphorylated neurofilament protein (NPNFP) labeled dysmorphic neurons in the grey matter exhibited accumulation of neurofilaments in the cytoplasm, abnormal cell soma enlargement and dendritic sprouting that are typical histopathological characteristics of the type II focal cortical dysplasia (FCD) in drug-resistant epilepsy [15,21,22,23,24]. Intriguingly, we found the percentage of NPNFP-expressing interstitial neurons was significantly increased in the white matter of FCD and TLE brain tissue compared with control brain tissue with no history of epilepsy (Control: 6.29 ± 3.39%; FCD: 37.26 ± 1.63%; TLE: 43.93 ± 16.62%. *n* = 3. Data are presented as the mean ± SEM. *p* < 0.05; one-way ANOVA with Tukey’s multiple comparisons test). We found no differences between drug-resistant epilepsy patients with FCD and TLE (*p =* 0.8527) (Figure 2A–D and Appendix A). Increased NPNFP expression in the interstitial neurons could be associated with the abnormal cellular development of interstitial neurons. Indeed, we found that the surface area of interstitial neurons in the white matter of FCD was significantly larger than that of TLE and control brain samples (Control: 144.09 ± 45.57 μm^2^, *n* = 74 from 6 samples, *p =* 0.4406; FCD: 321.67 ± 115.69 μm^2^, *n* = 56 from 3 samples, *p* < 0.0001; TLE: 185.21 ± 54.45 μm^2^, *n* = 22 from 3 samples, *p* < 0.001; one-way ANOVA with Tukey’s multiple comparisons test) (Figure 2E). These results suggested that the increased percentage of NPNFP-expressing interstitial neurons in the white matter could be associated with the pathology of the white matter from drug-resistant epilepsy patients.

### 3.3. Increased GABAergic and Glutamatergic Synaptic Density in the White Matter of Drug-Resistant Epilepsy Patients

Next, we examined whether the interstitial neurons in the white matter could receive synaptic inputs by immunostaining with GABAergic presynaptic marker, vesicular GABA transporter (VGAT), and glutamatergic presynaptic marker vesicular glutamate transporter 1 (VGLUT1). We found expression of both VGAT and VGLUT1 synaptic puncta in the white matter of human brain tissue (Figure 3 and Figure 4). Intriguingly, VGAT synaptic puncta were distributed in the peri-soma and dendrites of NPNFP-expressing interstitial neurons in the white matter of drug-resistant epilepsy patients (Figure 3A). We also found a significant increase in the density of VGAT-positive synaptic puncta in both the superficial white matter (SWM) of FCD (Control: 103.91 ± 38.15 per 10^4^ μm^2^, *n* = 6; FCD: 257.11 ± 19.44 per 10^4^ μm^2^, *n* = 3, *p* < 0.05; TLE: 194.55 ± 38.19 per 10^4^ μm^2^, *n* = 3. Control vs. TLE, *p* = 0.0938; FCD vs. TLE, *p =* 0.6722) and the deep white matter (DWM) of FCD and TLE compared with control brain samples (Control: 59.68 ± 10.37 per 10^4^ μm^2^, *n* = 6, *p* < 0.01; FCD: 204.48 ± 20.24 per 10^4^ μm^2^, *n* = 3, *p* < 0.001; TLE: 160.10 ± 23.66 per 10^4^ μm^2^, *n* = 3, *p* = 0.1965; one-way ANOVA with Tukey’s multiple comparisons test) (Figure 3A–C and Appendix A). In addition, the number of VGAT-positive puncta on the peri-soma region of the interstitial neurons of brain tissue sections from FCD and TLE was significantly higher than that in control brain samples (Control: 5.66. ± 1.64 per 100 μm^2^, *n* = 38 from 6 samples, *p* < 0.05; FCD: 8.76 ± 2.62 per 100 μm^2^, *n* = 32 from 3 samples, *p* < 0.01; TLE: 9.29 ± 2.42 per 100 μm^2^, *n* = 12 from 3 samples, *p =* 0.9103; one-way ANOVA with Tukey’s multiple comparisons test) (Figure 3A,D and Appendix A). Furthermore, we found that the number of VGAT-positive puncta distributed in the peri-soma of the NPNFP-positive interstitial neurons of brain tissue sections from FCD and TLE were significantly higher than that in NPNFP-negative interstitial neurons (NeuN^+^ NPNFP^+^: 29.50 ± 9.69, *n* = 16 cells from 6 samples; NeuN^+^ NPNFP^−^: 14.78 ± 4.66, *n* = 9 cells from 6 samples. *p* < 0.05. Unpaired two-tailed Student’s *t*-test). (Figure 3A,E and Appendix A).

We next quantified the expression of the glutamatergic synaptic marker VGLUT1 in the interstitial neurons in the white matter. We found an increased density of VGLUT1-positive synaptic puncta in both the superficial white matter (SWM) (Control: 93.47 ± 30.17 per 10^4^ μm^2^, *n* = 6, *p* < 0.0001; FCD: 343.02 ± 14.70 per 10^4^ μm^2^, *n* = 3, *p* < 0.0001; TLE: 334.94 ± 22.91 per 10^4^ μm^2^, *n* = 3, *p =* 0.9736) and the deep white matter (DWM) of FCD and TLE in comparison with control brain samples (Control: 57.32 ± 15.87 per 10^4^ μm^2^, *n* = 6, *p* < 0.0001; FCD: 215.58 ± 22.81 per 10^4^ μm^2^, *n* = 3, *p* < 0.0001; TLE: 253.62 ± 4.16 per 10^4^ μm^2^, *n* = 3, *p =* 0.2680; one-way ANOVA with Tukey’s multiple comparisons test) (Figure 4A–C and Appendix A). The number of VGLUT1-positive puncta in the peri-soma region of interstitial neurons of brain tissue sections from FCD and TLE were significantly higher than that from control brain samples (Control: 1.46 ± 0.84 per 100 μm^2^, *n* = 36 from 6 samples, *p* < 0.0001; FCD: 6.75 ± 2.06 per 100 μm^2^, *n* = 24 from 3 samples, *p* < 0.0001; TLE: 8.84 ± 1.86 per 100 μm^2^, *n* = 10 from 3 samples, *p =* 0.0937. One-way ANOVA with Tukey’s multiple comparisons test) (Figure 4A,D and Appendix A). We also found the number of VGLUT1-positive puncta distributed in the peri-soma of NPNFP-positive interstitial neurons of brain tissue sections from FCD and TLE showed no difference in comparison with NPNFP-negative interstitial neurons (NeuN^+^ NPNFP^+^: 21.38 ± 13.11, *n* = 13 cells from 6 samples; NeuN^+^ NPNFP^−^: 16.89 ± 3.81, *n* = 10 cells from 6 samples. *p* = 0.4229. Unpaired t-test) (Figure 4A,E and Appendix A). These results revealed the increased synaptic density of glutamatergic and GABAergic synapses in the white matter of drug-resistant epilepsy patients.

Taken together, our results showed the expression of cortical neuronal markers in interstitial neurons in the white matter of the adult human brain, suggesting a possible origin of interstitial neurons from cortical progenitor cells. We also found increased expression of NPNFP in interstitial neurons that exhibited dysmorphic cell morphology in the white matter of drug-resistant epilepsy patients, suggesting the aberrant cellular development of these interstitial neurons that could be associated with the pathology of drug-resistant epilepsy. In addition, we found that glutamatergic and GABAergic synaptic puncta were distributed in the peri-soma and primary dendrites of interstitial neurons. The density of glutamatergic and GABAergic synaptic puncta was increased in brain tissues from epilepsy patients with focal cortical dysplasia and temporal lobe epilepsy in comparison to brain tissues with no history of epilepsy. These results suggested that increased synaptic density of NPNFP-expressing interstitial neurons in the white matter could contribute to the epileptic network activity in drug-resistant epilepsy.

## 4. Discussion

Interstitial neurons identified in the white matter of adult human and non-human primate brain expressed various GABA and glutamatergic neuronal markers [2,9,25,26,27,28]. Interstitial neurons were considered to be remnant of subplate neurons that survived after apoptosis in the white matter during postnatal brain development [29]. Embryonic birth-dating lineage tracing in monkeys and carnivores suggested that interstitial neurons shared the same developmental origin with subplate neurons and deep-layer cortical neurons [11,14,15]. The expression of cortical layer markers CUX2, TLE4, and FOXP1 were identified in the embryonic human subplate [30], these neurons were considered as immature migratory neurons that co-expressed with DCX [31]. However, whether interstitial neurons in the adult human brain expressed cortical neuronal markers remains unexplored. In the present study, we demonstrated that a substantial portion of interstitial neurons in the white matter of human brain expressed cortical layer markers including upper-layer marker Satb2 and deep-layer markers Ctip2 and Tle4 (Figure 1 and Appendix A) [17,19]. These results supported the hypothesis that a subset of interstitial neurons could be originated from cortical neuron progenitor cells. The expression of upper-layer and deep-layer cortical neuronal markers suggested that interstitial neurons in the white matter could possibly be derived from cortical neurons that were not able to migrate to the cortex during cortical neurogenesis.

An increased number of heterotopic interstitial neurons was identified in the temporal lobe as compared to frontal or occipital cortical white matter [3,32]. Additionally, the number or distribution of interstitial neurons in the white matter could be affected by prenatal lesion and were also implicated to be associated with neurological or neuropsychiatric diseases such as epilepsy and schizophrenia [29,33,34,35,36]. Disrupted cortical lamina or abnormal distribution of cortical pyramidal neurons were associated with seizure genesis in drug-resistant epilepsy patients with focal cortical dysplasia [15]. The proportion of upper and deep cortical layer markers’ expression in interstitial neurons could originate from aberrant cellular development from cortical neuron progenitors. The NPNFP-expressing interstitial neurons exhibited enlarged cell soma in FCD and TLE brain samples, suggesting possible morphological dystrophy of interstitial neurons that could be associated with the pathology of the white matter from drug-resistant epilepsy patients (Figure 2). In addition, linear regression and correlation analysis showed no correlation between the number of NeuN- or NPNFP-positive interstitial neurons or the number of VGAT- or VGLUT1-positive puncta with the age of brain samples (Figure 5).

The physiological functions of interstitial neurons in the adult brain are not fully understood. The dendrites of interstitial neurons could extend to layer IV of the cortex and receive synaptic inputs from the cortex [7,37,38]. A few studies have characterized the action potential firing properties and synaptic connectivity of interstitial neurons in the white matter of rodent brain [39,40]. Interstitial neurons could receive glutamatergic and GABAergic synaptic inputs from the deep-layer cortex, hippocampus, or striatum. The axonal branches of interstitial neurons were mainly distributed in the deep-layer cortex and could form functional synapses with deep-layer cortical neurons [41]. These results demonstrate the functional integration of interstitial neurons into the cortical networks in the rodent brain. We found increased glutamatergic and GABAergic synaptic density in the peri-soma and primary dendrites of NPNFP-expressing interstitial neurons in the white matter of drug-resistant epilepsy patients when compared with control brain tissue. The imbalance of synaptic excitation and the inhibition of dysmorphic neurons in the grey matter could contribute to seizure genesis in FCD [21,42,43]. Therefore, the increased glutamatergic and GABAergic synaptic density of interstitial neurons in the white matter could be integrated into the epileptic neuronal networks and contribute to seizure genesis in drug-resistant epilepsy. Consistent with our findings, a recent study showed that the synaptic density in the neocortical white matter of TLE patients was significantly higher than that of controls, and the synaptic density in the white matter was significantly correlated with the postoperative prognosis of drug-resistant epilepsy patients with TLE [14]. Together, these results suggested that interstitial neurons in the white matter of the adult human brain could potentially participate in the epileptic network and contribute to seizure genesis in drug-resistant epilepsy.

## 5. Conclusions

Our study identified the expression of typical cortical markers Satb2, Ctip2, and Tle4 in interstitial neurons in the white matter of the adult human brain, supporting the hypothesis that a substantial portion of interstitial neurons could originate from cortical neuron progenitor cells. In addition, we found a significant increase in both NPNFP expression and the density of GABA and glutamate synapses on interstitial neurons in the white matter of drug-resistant epilepsy patients compared with control brain tissue with no history of epilepsy. These results suggested that interstitial neurons could contribute to the epileptic network in drug-resistant epilepsy patients.

## 6. Limitations and Future Directions

The control brain samples were from postmortem brains and did not strictly match the brain regions or ages of postoperative brain samples from drug-resistant epilepsy patients. We have performed strict control experiments to ensure that all the brain samples used in our study were qualified for immunostaining experiments, including the expression and distribution of canonical cortical layer markers and NeuN in cortical neurons in the gray matter. Future studies using appropriate model systems to investigate the developmental origin, functional synaptic connections and roles of interstitial neurons in epilepsy are warranted.

## Figures and Tables

**Figure 1 brainsci-13-00626-f001:**
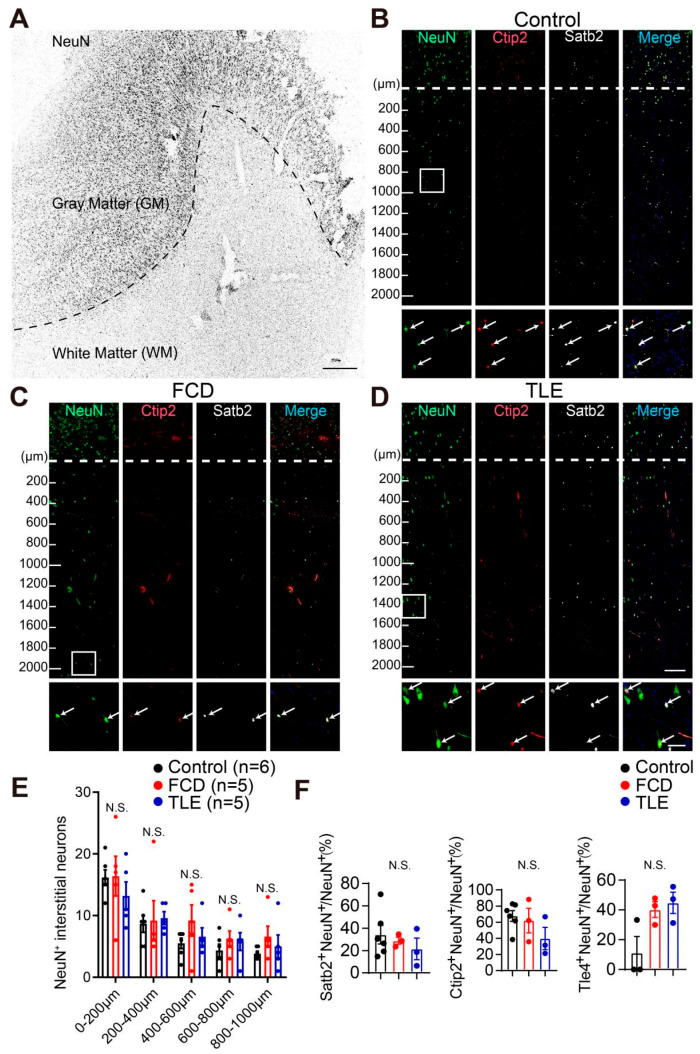
The distribution of cortical layer markers in interstitial neurons in the white matter of brain tissues from FCD, TLE, and control. (**A**) Tile scan image showing the distribution of NeuN-positive neurons in the gray matter and white matter of the frontal lobe of adult control human brain. A dashed line indicates the border between gray matter and white matter. The rectangular box indicates the image acquisition area in panel (**B**). Scale bar, 1000 μm. (**B**) Tile scan images showing the distribution of interstitial neurons expressing cortical layer markers in the white matter of brain tissues from control, FCD (**C**), and TLE (**D**). Arrows indicated co-expression of cortical layer markers with NeuN. NeuN, neuronal marker; Satb2, upper-layer cortical pyramidal neuron marker; Ctip2, deep-layer cortical pyramidal neuron marker. Scale bar, 200 μm in the original image (up) and 50 μm in zoom-in image (bottom). A dashed line indicates the border between gray matter and white matter. (**E**) Statistics showing the number of NeuN-positive interstitial neurons quantified at different subregions from the superficial to deep white matter of brain tissues from control (*n* = 6), FCD (*n* = 5), and TLE (*n* = 5) (every 200 μm bin from the border of gray matter and white matter to deep white matter; 0–200 μm, p = 0.5647; 200–400 μm, p = 0.9467; 400–600 μm, p = 0.3005; 600–800 μm, p = 0.5346; 800–1000 μm, p = 0.3847; one-way ANOVA with Tukey’s multiple comparisons test). (**F**) Statistics showing the proportion of Satb2-positive (SD: control = 20.42, FCD = 5.49, and TLE = 16.73), Ctip2-positive (SD: control = 16.73, FCD = 8.23, and TLE = 12.16), or Tle4-positive (SD: control = 21.65, FCD = 9.20, and TLE = 12.37) neurons in all of the interstitial neurons of brain tissues from control, FCD, and TLE. Data are presented as the mean ± SEM. N.S., no significance. Control: neocortical brain samples with no history of epilepsy; FCD: focal cortical dysplasia; TLE: temporal lobe epilepsy. SD: standard deviation.

**Figure 2 brainsci-13-00626-f002:**
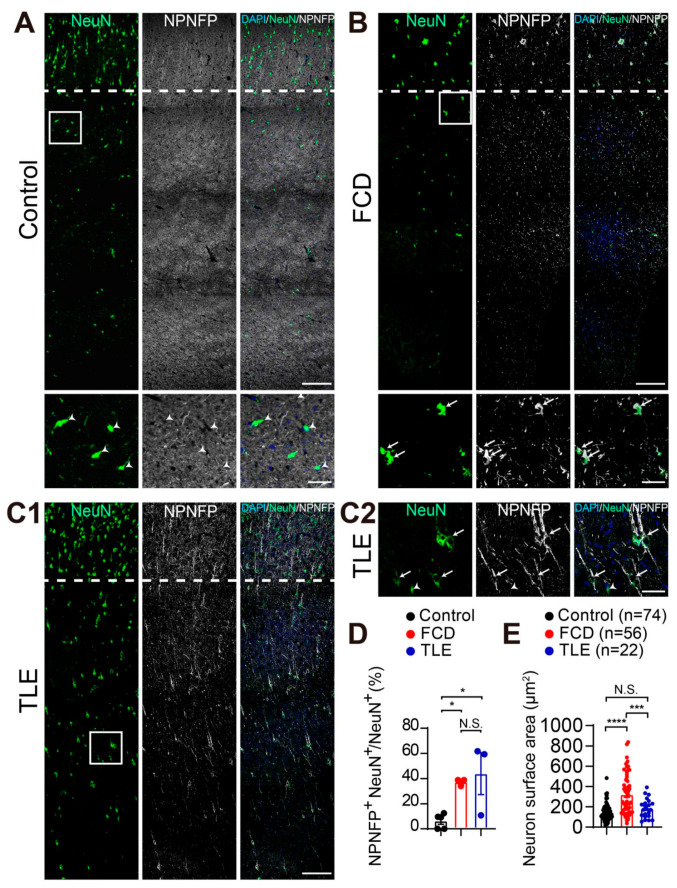
Increased NPNFP expression in interstitial neurons in the white matter of drug-resistant epilepsy patients. (**A**) Tile scan images and zoom-in images (bottom) showing the expression of NPNFP in interstitial neurons in the white matter of brain tissue sections from 3 independent samples of control, FCD (**B**), and TLE (**C1**, tile scan image; **C2**, zoom-in image). Arrows indicate the co-expression of NPNFP with NeuN. Arrowheads indicate that no NPNFP expression was detected in the interstitial neurons. Scale bar, 200 μm in the tile scan image (up) and 50 μm in zoom-in image (bottom). A dashed line indicates the border between gray matter and white matter. NPNFP: non-phosphorylated neurofilament protein marker, SMI311. (**D**) Statistics showing the proportion of NPNFP-positive neurons within the total interstitial neurons of brain tissue sections from control, FCD, and TLE (SD: control = 5.87, FCD = 2.82 and TLE = 28.79), respectively. Data are presented as the mean ± SEM. (**E**) Statistics showing the surface area of interstitial neurons in the white matter (SD: control = 78.92, FCD = 200.37 and TLE = 94.31). Data are presented as the mean ± SEM. **** *p* < 0.0001, *** *p* < 0.001, * *p* < 0.05, N.S., no significance; one-way ANOVA with Tukey’s multiple comparisons test.

**Figure 3 brainsci-13-00626-f003:**
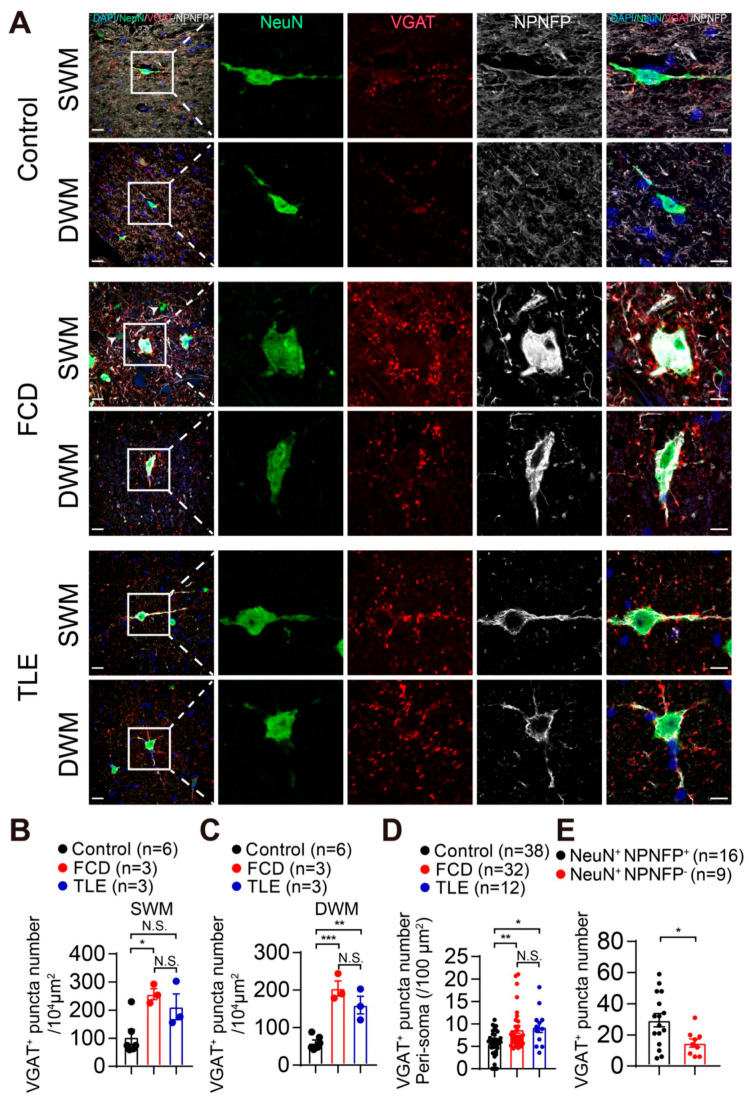
Increased GABAergic synaptic density in the white matter of drug-resistant epilepsy patients. (**A**) Example images showing the distribution of the GABAergic synaptic marker VGAT in the peri-soma and dendrites of interstitial neurons in the superficial and deep white matter of brain tissue sections from FCD, TLE, and control, respectively. Scale bar, 20 μm in the original image (left) and 10 μm in zoom-in image (right). SWM: superficial white matter; DWM: deep white matter. (**B**) Statistics showing the density of VGAT-positive puncta distributed in the superficial white matter (SWM) (SD: control = 66.08, FCD = 33.67 and TLE = 66.15) and in the deep white matter (DWM) (SD: control = 17.96, FCD = 35.06 and TLE = 40.98) (**C**) of FCD and TLE compared with control brain samples. (**D**) Statistics showing the number of VGAT-positive puncta distributed in the peri-soma and primary dendrites of interstitial neurons of brain tissue sections from control (*n* = 38 cells from 6 samples), FCD (*n* = 32 cells from 3 samples), and TLE (*n* = 12 cells from 3 samples) (SD: control = 2.84, FCD = 4.54 and TLE = 4.19), respectively. (**E**) Statistics showing the number of VGAT-positive puncta distributed in the peri-soma of the NeuN^+^ NPNFP^+^ and NeuN^+^ NPNFP^−^ interstitial neurons of brain tissue sections from FCD and TLE (SD: NeuN^+^ NPNFP^+^ = 16.78, NeuN^+^ NPNFP^−^ = 8.07). Data are presented as the mean ± SEM. *** *p* < 0.001, ** *p* < 0.01, * *p* < 0.05, N.S., no significance. One-way ANOVA with Tukey’s multiple comparisons test and unpaired Student’s *t*-test.

**Figure 4 brainsci-13-00626-f004:**
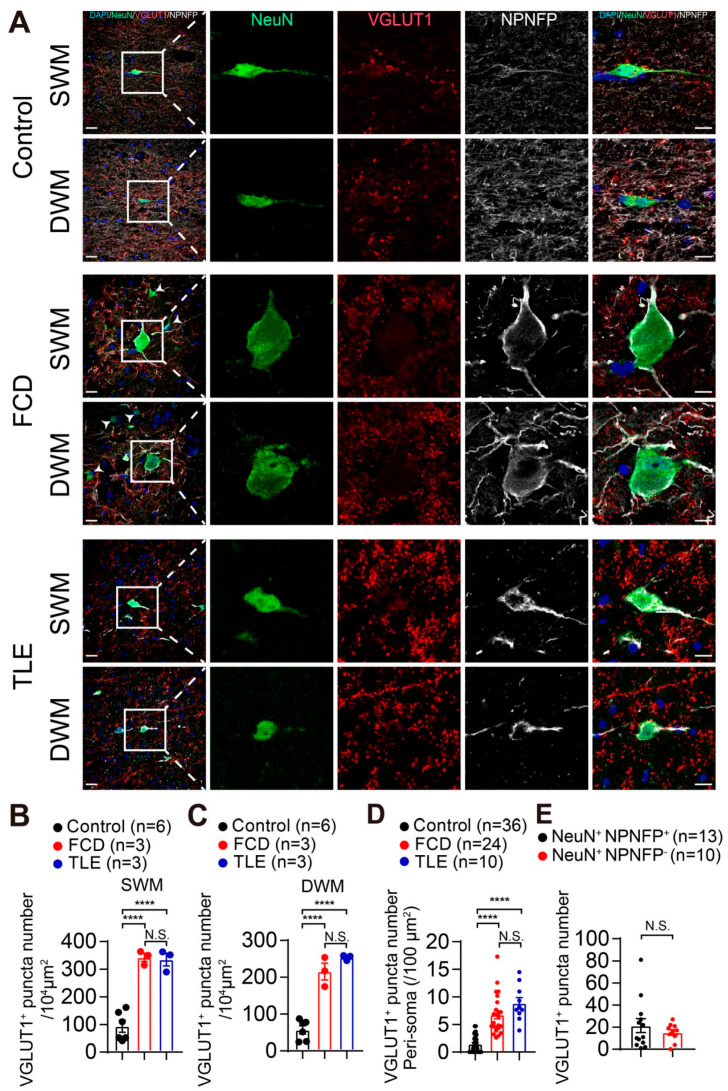
Increased glutamatergic synaptic density in the white matter of drug-resistant epilepsy patients. (**A**) Example images showing the distribution of VGLUT1 in the peri-soma and primary dendrites of interstitial neurons in the superficial and deep white matter of brain tissue sections from FCD, TLE and control, respectively. Scale bar, 20 μm in the original image (left) and 10 μm in zoom-in image (right). SWM: superficial white matter; DWM: deep white matter. (**B**) Statistics showing the density of VGLUT1 positive synaptic puncta in the superficial white matter (SWM) (SD: control = 52.26, FCD = 25.46 and TLE = 39.68) and in the deep white matter (DWM) (SD: control = 27.49, FCD = 39.51 and TLE = 7.21) (**C**) of FCD and TLE compared with control brain samples. (**D**) Statistics showing the number of VGLUT1 positive puncta distributed on the peri-soma of interstitial neurons of brain tissue sections from control (*n* = 36 cells from 6 samples), FCD (*n* = 24 cells from 3 samples) and TLE (*n* = 10 cells from 3 samples) (SD: control = 1.45, FCD = 3.57 and TLE = 3.22), respectively. (**E**). Statistics showing the number of VGLUT1 positive puncta distributed in the peri-soma of NeuN^+^ NPNFP^+^ and NeuN^+^ NPNFP^−^ interstitial neurons of brain tissue sections from FCD and TLE (SD: NeuN^+^ NPNFP^+^ = 22.71, NeuN^+^ NPNFP^−^ = 6.60). Data are presented as mean ± SEM. **** *p* < 0.0001, N.S., no significance. One-way ANOVA with Tukey’s multiple comparisons test.

**Figure 5 brainsci-13-00626-f005:**
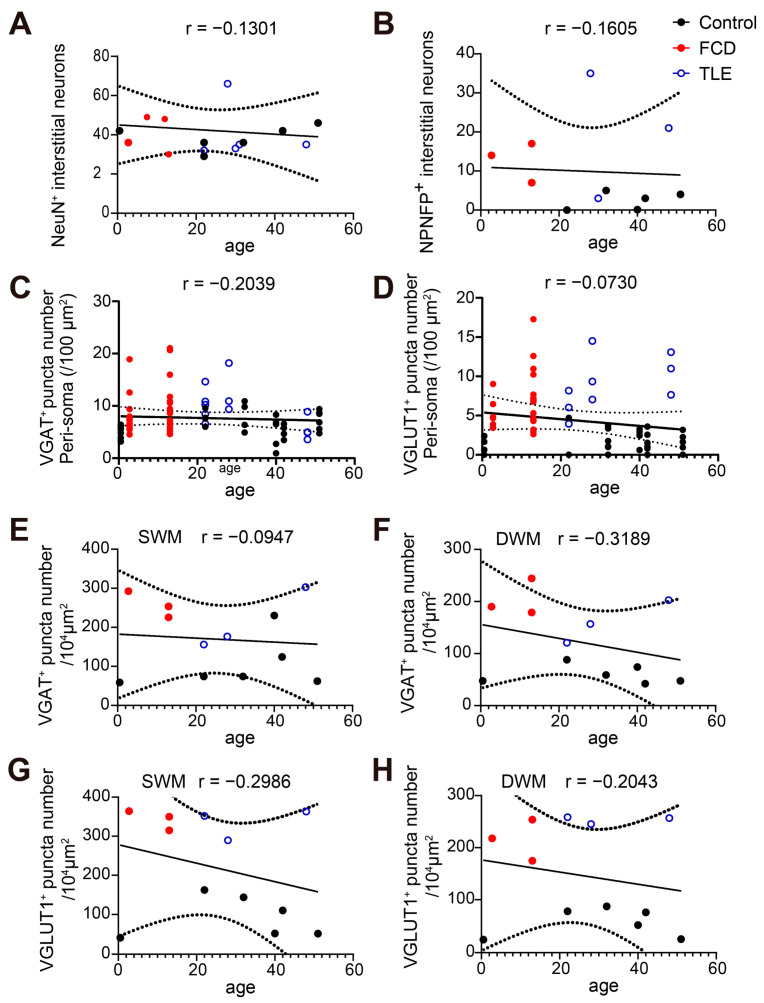
Linear regression and correlation analysis showed no correlation between the measured parameters and the age of brain samples. (**A**–**F**) Linear regression and correlation analysis showed no correlation between the number of NeuN- (**A**) or NPNFP-positive (**B**) interstitial neurons or the number of VGAT- (**C**,**E**,**F**) or VGLUT1-positive (**D**,**G**,**H**) puncta and the age of brain samples. SWM: superficial white matter; DWM: deep white matter. r: Pearson correlation coefficient.

**Table 1 brainsci-13-00626-t001:** Clinical data of the patients.

ID	Diagnosis	Age at Epilepsy Onset	Epilepsy Duration	Age at Surgery	Monthly Seizure Frequency	Site of Surgery
FCD 15	FCD IIB	9.25	10	13.75	*	R.gfs
FCD 21	FCD IIB	6.75	0.75	7.5	45	L.gt
FCD 24	FCD IIB	0.75	2	2.75	*	R.gf
FCD 25	FCD IIB	3	10	13	90	R.gfs
FCD 27	FCD IIB	11	5	16	45	L.gfs
TLE 31	Crypto	10	18	28	3	R.gt
TLE 32	Crypto	12	10	22	15.3	R.gt
TLE 36	Crypto	44	4	48	6	L.gt
TLE 37	HS	9	7	16	4	L.gt
TLE 40	HS	7.25	6	13.25	4.5	R.gt
TLE 41	HS	13.25	0.75	14	1.5	L.gt
**Autopsy**		**Age at Death**	**Postmortem Delay (min)**			
2017CBB047	Autopsy	0.5	253	/	/	R.gfs 3
2018CBB037	Autopsy	42	768	/	/	R.gfs 3
2019CBB013	Autopsy	51	473	/	/	R.gfs 3
2022CBB015	Autopsy	22	805	/	/	R.gfs 3
2020CBB017	Autopsy	40	621	/	/	R.gfs 3
2021CBB052	Autopsy	32	220	/	/	R.gfs 3

Abbreviations: FCD = focal cortical dysplasia; TLE = temporal lobe epilepsy; Crypto = cryptogenic; HS = hippocampal sclerosis; R = right; L = left; g = gyrus; t = temporal lobe; f = frontal lobe; s = superior; * clinical data not recorded/data not available.

## Data Availability

All of the relevant data generated and analyzed during the current study are available upon reasonable request to the corresponding author.

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
