# Peer review of "Cortical Layer Markers Expression and Increased Synaptic Density in Interstitial Neurons of the White Matter from Drug-Resistant Epilepsy Patients"

_brainsci, 2023, doi:10.3390/brainsci13040626_

Round 1

Reviewer 1 Report

In their manuscript entitled “Cortical layer markers expression and increased synaptic density in interstitial neurons of the white matter from drug-resistant epilepsy patients” the authors Jiachao Yang, Mi Wang, Yang Lyu, and Jiadong Chen investigated the expression of typical cortical markers Sathb2, Ctip2 and Tle4 in interstitial neurons of human samples from two types of epileptic patients and from a control group. The observed that all of these markers were expressed in interstitial neurons, with no significant differences between the two epileptic cohorts and the control samples. In addition, the authors investigated the number of GABAergic and glutamatergic synapses, quantified by VGAT and VGLUT1 expression, in these samples. They observed a significant higher density of GABA and glutamate synapses on interstitial neurons in both of the epileptic cohorts. They conclude from these observations that a substantial portion of interstitial neurons could originate from cortical neuron progenitor cells. The topic of the manuscript is interesting, the methods are sound, the results reliable, adequately displayed and in most parts fairly discussed. There are only a few points that should be considered to improve the manuscript.

Points:

 1.) Line 336ff: In the whole discussion I missed the point, whether your results can discriminate if the interstitial neurons are remnants of subplate cells or of "classical" cortical neurons ought to populate the cortical layers II to VI. Therefore, you should discuss whether your used cortical markers, Satb2, Ctip2 or Tle4, are expressed in subplate neurons. See e.g. publications by Molnar or recent articles from Ivica Kostovic's group. Please reconsider in this respect also the initial statement of your abstract (Line 13-14) “A subset of interstitial neurons express molecular markers of subplate neurons, but whether interstitial neurons express cortical layer markers remains unexplored.”

2.) Line 231-234: “the number of VGAT positive puncta on the peri-soma region of interstitial neurons of brain tissue sections”. Actually, I cannot follow how you obtained this number. In the Materials you stated that the density of synaptic puctae was "normalized to neuron surface area. Do you calculate the surface area from 3D reconstructions of neurons with the Imaris software? Please describe the procedures in more detail.

3.) p-values. I would suggest that you provide the p-values (as you did in the section 3.1) instead of the p-limits (“p>0.05”) also for sections 3.2-3.3.

4.) Lines 281-285: “In addition, we quantified the surface area …. Figure 4E). In my opinion this important information should be brought separately and much earlier, e.g. within section 3.1 or at the beginning of this section. At least, this information is not restricted to the VGLUT1 synapses. In this respect, I wonder why you did not add the surface areas determined in the VGAT staining to these numbers.

Line 33: Consider to provide here also a reference to cortical interstitial neurons.

Line 43: There are also hints that remnant subplate neurons contribute to Layer VIb.

Legend to Figures, e.g. Line 185: You may consider to remove the detailed statistical information from the legend, as the whole information is also provided in the main text of the results section.

Line 200, Fig. 2a-c: The differences in the fibrillary NPNFP staining in Fig. 2A-c are intriguing. Do you consider that this is due to differences in the NPNFP expression between the groups?

Line 204: “Increased NPNFP expression in the interstitial neurons was associated with abnormal cellular development of interstitial neurons”. This relation is not provided from your results, please support with a figure/analysis. If this statement is based on published results, please provide a reference.

Lines 222-247: You may consider here to organize the results for the vGAT data as for the vGLUT results: providing first the general density in the sWM and the dWM and subsequently the synaptic density at the surface of interstitial neurons. Please note in this respect that also in the legend to Fig. 4 the sequence of panes B-D does not match the presentation of the experiments in the main text.

Line 238 and many other cases: Please correct 104 to 104 throughout the manuscript.

Line 276: Please use “VGLUT1” consistently.

Author Response

Response to Reviewer 1's comments

In their manuscript entitled “Cortical layer markers expression and increased synaptic density in interstitial neurons of the white matter from drug-resistant epilepsy patients” the authors Jiachao Yang, Mi Wang, Yang Lyu, and Jiadong Chen investigated the expression of typical cortical markers Sathb2, Ctip2 and Tle4 in interstitial neurons of human samples from two types of epileptic patients and from a control group. The observed that all of these markers were expressed in interstitial neurons, with no significant differences between the two epileptic cohorts and the control samples. In addition, the authors investigated the number of GABAergic and glutamatergic synapses, quantified by VGAT and VGLUT1 expression, in these samples. They observed a significant higher density of GABA and glutamate synapses on interstitial neurons in both of the epileptic cohorts. They conclude from these observations that a substantial portion of interstitial neurons could originate from cortical neuron progenitor cells. The topic of the manuscript is interesting, the methods are sound, the results reliable, adequately displayed and in most parts fairly discussed. There are only a few points that should be considered to improve the manuscript.

Points:

 1.) Line 336ff: In the whole discussion I missed the point, whether your results can discriminate if the interstitial neurons are remnants of subplate cells or of "classical" cortical neurons ought to populate the cortical layers II to VI. Therefore, you should discuss whether your used cortical markers, Satb2, Ctip2 or Tle4, are expressed in subplate neurons. See e.g. publications by Molnar or recent articles from Ivica Kostovic's group. Please reconsider in this respect also the initial statement of your abstract (Line 13-14) “A subset of interstitial neurons express molecular markers of subplate neurons, but whether interstitial neurons express cortical layer markers remains unexplored.”

We thank the reviewer for suggestion. In the postnatal rodent brain, a portion of subplate neurons become interstitial neurons and layer VIb neurons labeled by Cplx3, Nurr1, Lpar1 and CTGF. In the embryonic human brain, the expression of cortical layer markers CUX2, TLE4, and FOXP1 were identified in the embryonic human subplate (Miskic et al., Cells. 2021), these neurons were considered as immature migratory neurons that co-expressed DCX (Kubo et al., JCI. Insight. 2017). Our findings showed that Satb2, Ctip2 and TLE4 are expressed in interstitial neurons in the white matter of adult human brain, supporting the hypothesis that interstitial neurons could originate from cortical neuron progenitor cells. We have revised the abstract and discussion accordingly (Line 15 and Line 342-344).

2.) Line 231-234: “the number of VGAT positive puncta on the peri-soma region of interstitial neurons of brain tissue sections”. Actually, I cannot follow how you obtained this number. In the Materials you stated that the density of synaptic puctae was "normalized to neuron surface area. Do you calculate the surface area from 3D reconstructions of neurons with the Imaris software? Please describe the procedures in more detail.

The number of VGAT and VGLUT1 positive synaptic puncta were quantified from ROI (100 × 100 μm2) image of each brain sample in the superficial (200-500 μm from the border of gray and white matter) or deep white matter (500-1000 μm from the border of gray and white matter), respectively. The neuron surface area was measured by cell outline sketch of the soma region labeled by NeuN immunostaining and area measurement using ImageJ software. The number of synaptic puncta in the peri-soma region of interstitial neurons was quantified by averaging the number of synaptic puncta divided by neuron surface area (per 100 μm2) from at least 6 interstitial neurons from each brain sample. We described the detailed procedures in revised methods accordingly. (Line 138-146)

3.) p-values. I would suggest that you provide the p-values (as you did in the section 3.1) instead of the p-limits (“p>0.05”) also for sections 3.2-3.3.

We thank the reviewer for suggestion and provided the p-values for (“p>0.05”) in the sections 3.2-3.3 in the revised manuscript.

4.) Lines 281-285: “In addition, we quantified the surface area …. Figure 4E). In my opinion this important information should be brought separately and much earlier, e.g. within section 3.1 or at the beginning of this section. At least, this information is not restricted to the VGLUT1 synapses. In this respect, I wonder why you did not add the surface areas determined in the VGAT staining to these numbers.

We thank the reviewer for suggestion. We moved the quantification of surface area of interstitial neurons to Figure 2E in section 3.2 in revised manuscript. The quantification of neuronal surface area included neurons from both VGLUT1 and VGAT immunostaining.

5.) Line 33: Consider to provide here also a reference to cortical interstitial neurons.

We revised the manuscript according to the reviewer's suggestion with additional references (Kostovic and Rakic J. Neurocytol. 1980; Judas et al., J. Anat. 2010) (Line 34 and Line 465-468). 

6.) Line 43: There are also hints that remnant subplate neurons contribute to Layer VIb.

We revised the manuscript according to the reviewer's suggestion and included additional references (Hoerder-Suabedissen et al., Cereb. Cortex 2009; Hoerder-Suabedissen et al., Cereb. Cortex 2013) (Line 45 and Line 484-488).

7.) Legend to Figures, e.g. Line 185: You may consider to remove the detailed statistical information from the legend, as the whole information is also provided in the main text of the results section.

We thank the reviewer for suggestion and revised the legends of Figures 2,3 and 4 accordingly (Line 198,236,272,274,278,279,311,313,316).

8.) Line 200, Fig. 2a-c: The differences in the fibrillary NPNFP staining in Fig. 2A-c are intriguing. Do you consider that this is due to differences in the NPNFP expression between the groups?

The NPNFP could also be expressed in axons in brain injury (Johnson et al., Acta Neuropathol. 2016; Herrero-Herranz et al., Neurobiol Dis. 2008). Therefore the fibrillay NPNFP staining could potentially represent axonal injury in the white matter of drug-resistant epilepsy. However, we found variability of fibrillary NPNFP staining between different samples within the same group (Supplemental figure 2). Therefore we were not able to quantify the differences of NPNFP expression between the groups. 

9.) Line 204: “Increased NPNFP expression in the interstitial neurons was associated with abnormal cellular development of interstitial neurons”. This relation is not provided from your results, please support with a figure/analysis. If this statement is based on published results, please provide a reference.

We thank the reviewer for suggestion. The NPNFP expression in the grey matter is considered as a marker for dysmorphic neurons in drug-resistant epilepsy with cortical dysplasia (Ljungberg et al., Ann. Neurol. 2006; Rossini et al., Epilepsy Research 2017; Rossini et al. Acta Neuropathologica Communications 2014). In addition, we quantified the soma size of interstitial neurons and found the surface area of interstitial neurons in the white matter of FCD was significantly larger than that of TLE and control brain samples. We have revised the manuscript with supporting analysis and references accordingly (Line 214-219 and Line 510-518).

10.) Lines 222-247: You may consider here to organize the results for the vGAT data as for the vGLUT results: providing first the general density in the sWM and the dWM and subsequently the synaptic density at the surface of interstitial neurons. Please note in this respect that also in the legend to Fig. 4 the sequence of panes B-D does not match the presentation of the experiments in the main text.

We thank the reviewer for suggestion and revised the sequence of panels B-D in Figure 3 and Figure 4 accordingly.

11.) Line 238 and many other cases: Please correct 104 to 104 throughout the manuscript.

We thank the reviewer for suggestion and revised the manuscript accordingly.

12.) Line 276: Please use “VGLUT1” consistently.

We thank the reviewer for suggestion and revised the manuscript accordingly (Line 436).

Reviewer 2 Report

The authors provide evidence of expression of cortical layer markers in the interstitial neurons in the white matter of adult human brain. Their data supports the hypothesis that a subset of interstitial neurons could be originated from cortical neuron progenitor cells. Furthermore, they report that both glutamatergic and GABAergic synaptic puncta increased in drug resistant epilepsy patients compared to those with no history of epilepsy, indicating that interstitial neurons in the white matter of adult human brain could contribute to the genesis of seizures and be a part of epileptic network in the drug resistant epileptic patients.

Comments:

Overall this is a well written manuscript. Major strength of the study is that it uses human samples and compare between both conditions using imaging.

The weakness of the study is that there are no loss of function or gain of function experiments to absolutely confirm the findings. 

Author Response

Response to Reviewer 2's comments

The authors provide evidence of expression of cortical layer markers in the interstitial neurons in the white matter of adult human brain. Their data supports the hypothesis that a subset of interstitial neurons could be originated from cortical neuron progenitor cells. Furthermore, they report that both glutamatergic and GABAergic synaptic puncta increased in drug resistant epilepsy patients compared to those with no history of epilepsy, indicating that interstitial neurons in the white matter of adult human brain could contribute to the genesis of seizures and be a part of epileptic network in the drug resistant epileptic patients.

Comments:

Overall this is a well written manuscript. Major strength of the study is that it uses human samples and compare between both conditions using imaging.

We thank the reviewer for positive remarks on our manuscript. 

The weakness of the study is that there are no loss of function or gain of function experiments to absolutely confirm the findings. 

We agree with the reviewer. We did not demonstrate whether the increase of glutamatergic and GABAergic synapses in interstitial neurons could be sufficient to induce seizure activity. Future studies using appropriate model system to investigate the origin of interstitial neurons as well as the roles in epilepsy are warranted. We have added a segment discussing the limitations of our study (Line 409-417).

Reviewer 3 Report

The authors investigated the interstitial neurons’ developmental origin and found that a subset of interstitial neurons could be originated from cortical neuron progenitor cells.  On the knowledge of the number or distribution of interstitial neurons in the white matter could be associated with neurological or neuropsychiatric diseases such as epilepsy and schizophrenia, the authors then analyzed the molecular markers and synaptic density of interstitial neurons in the white matter of drug-resistant epilepsy patients as well as control brain samples with no history of epilepsy. The study found significant differences in both biomarker NPNFP expression and GABAergic and glutamatergic synaptic density between drug-resistant epilepsy patients and the control brain tissues with no history of epilepsy and concluded that interstitial neurons in the white matter of the adult human brain could potentially participate in the epileptic network and contribute to seizure genesis in drug-resistant epilepsy. The paper is well-written and organized. The experiments were designed and conducted reasonably and the results strongly support the conclusion.  

Author Response

Response to Reviewer 3's comments

The authors investigated the interstitial neurons’ developmental origin and found that a subset of interstitial neurons could be originated from cortical neuron progenitor cells.  On the knowledge of the number or distribution of interstitial neurons in the white matter could be associated with neurological or neuropsychiatric diseases such as epilepsy and schizophrenia, the authors then analyzed the molecular markers and synaptic density of interstitial neurons in the white matter of drug-resistant epilepsy patients as well as control brain samples with no history of epilepsy. The study found significant differences in both biomarker NPNFP expression and GABAergic and glutamatergic synaptic density between drug-resistant epilepsy patients and the control brain tissues with no history of epilepsy and concluded that interstitial neurons in the white matter of the adult human brain could potentially participate in the epileptic network and contribute to seizure genesis in drug-resistant epilepsy. The paper is well-written and organized. The experiments were designed and conducted reasonably and the results strongly support the conclusion.  

We thank the reviewer for positive remarks on our manuscript.

Reviewer 4 Report

Overall, this is a competently written manuscript on a topic that deserves more recognition in the field of neuroscience. However, I believe the authors need to expand on the background of their work (especially in the introduction section) and decribe the methodology (and its limitations) in more detail. Please find more specific comments below.

Introduction

lines 43 - 45: many interstitial neurons also express somatostatin and NPY - very important GABAergic markers - these same GABAergic neurons typically also express MAP2 (a marker often associated with projection neurons); see  Banovac et al. 2022 (The Distinct Characteristics of Somatostatin Neurons in the Human Brain) and Sedmak and Judaš 2021 (White Matter Interstitial Neurons in the Adult Human Brain)

lines 45 - 48: a citation is needed to support this statement

Materials and methods

The cortical regions that were analyzed need to be described more precisely (using cytoarchitectonic nomenclature - e.g. Brodmann areas) and the authors need to give more details on how these regions were delineated (especially for control brains where it is important that the region corresponds to that of the post-operative tissue).

Since in some images the authors show the deep and superficial white matter, it would be important to describe how these were delineated from each other.

Table 1 - postmortem delay is missing for control samples

Besides presenting the data as mean +/- SEM, the authors should also give the values for the standard deviation as this gives more insight into the variability of the dataset. SEM gives more insight in how precise the mean can be determined from the given dataset.

Results

lines 148 - 151: the finding that interstitial neurons express cortical layer markers is important, but not necessarily completely unexpected given their developmental orgin - consider phrasing the sentence in a different manner (e.g. "it is important to note" instead of "intriguingly")

Discussion

A segment discussing the limitations of the study should be added (e.g. the authors are comparing postmortem controls to postoperative tissue - the authors need to discuss how this could affect the stainings they performed and their subsequent analyses).

Supplementary figure 4 (description) - VGLUT1 is a glutamatergic marker, not GABAergic

Author Response

Response to Reviewer 4's comments

Overall, this is a competently written manuscript on a topic that deserves more recognition in the field of neuroscience. However, I believe the authors need to expand on the background of their work (especially in the introduction section) and decribe the methodology (and its limitations) in more detail. Please find more specific comments below.

Introduction

  1. lines 43 - 45: many interstitial neurons also express somatostatin and NPY - very important GABAergic markers - these same GABAergic neurons typically also express MAP2 (a marker often associated with projection neurons); see  Banovac et al. 2022 (The Distinct Characteristics of Somatostatin Neurons in the Human Brain) and Sedmak and Judaš 2021 (White Matter Interstitial Neurons in the Adult Human Brain)

We revised the text and included additional references following the reviewer's suggestions in the revised manuscript. (Line 46,47,469,489)

  1. lines 45 - 48: a citation is needed to support this statement

We added additional references (Garcia-Marin et al J. Comp. Neurol. 2010; Meyer et al., Exp Brain Res. 1992) following the reviewer's suggestions in the revised manuscript. (Line 475,491)

Materials and methods

  1. The cortical regions that were analyzed need to be described more precisely (using cytoarchitectonic nomenclature - e.g. Brodmann areas) and the authors need to give more details on how these regions were delineated (especially for control brains where it is important that the region corresponds to that of the post-operative tissue).

We thank the reviewer for suggestion. We agree with the reviewer that it is important to use brain regions from control brain sample corresponding to that of the post-operative brain tissue from drug-resistant epilepsy patients. The control brain tissue were collected from superior frontal gyrus that can be mapped to relevant Broadmann areas. However, we are not able to map the precise Broadmann areas of the postsurgical brain tissue that were mostly from frontal lobe or temporal lobe. We have described the cortical regions of human brain tissue used in our study in more detail in revised Table 1. We have also discussed the limitations of our study in the revised manuscript (Line 87,409).

  1. Since in some images the authors show the deep and superficial white matter, it would be important to describe how these were delineated from each other.

We thank the reviewer for suggestion. The border of gray matter and white matter was determined by the distribution of NeuN positive neurons and DAPI counterstain. The superficial white matter (SWM) region include brain region that was 200-500 μm away from the border of gray and white matter, while the deep white matter (DWM) region include brain region that was 500-1000 μm away from the border of gray and white matter. We have revised the methods accordingly (Line 137-141).

  1. Table 1 - postmortem delay is missing for control samples

We added the postmortem delay for control samples in revised Table 1 following the reviewer's suggestions.

  1. Besides presenting the data as mean +/- SEM, the authors should also give the values for the standard deviation as this gives more insight into the variability of the dataset. SEM gives more insight in how precise the mean can be determined from the given dataset.

We thank the reviewer for suggestion and added the values for the standard deviation in the revised figure legends accordingly.

Results

  1. lines 148 - 151: the finding that interstitial neurons express cortical layer markers is important, but not necessarily completely unexpected given their developmental orgin - consider phrasing the sentence in a different manner (e.g. "it is important to note" instead of "intriguingly")

We thank the reviewer for suggestion and revised the manuscript accordingly (Line 159).

Discussion

  1. A segment discussing the limitations of the study should be added (e.g. the authors are comparing postmortem controls to postoperative tissue - the authors need to discuss how this could affect the stainings they performed and their subsequent analyses).

We thank the reviewer for suggestion. The control brain samples were from postmortem brains and did not strictly match the brain regions or ages of postoperative brain samples from drug-resistant epilepsy patients. We have performed strict control experiments to ensure all the brain samples used in our study were qualified for immunostaining experiments, including the expression and distribution of canonical cortical layer markers and NeuN in cortical neurons in the grey matter. We have added a segment discussing the limitations of our study (Line 409-417).

  1. Supplementary figure 4 (description) - VGLUT1 is a glutamatergic marker, not GABAergic

We thank the reviewer for suggestion and revised the figure legend of supplementary figure 4 accordingly.
